# Exploring the Rural Revitalization Effect under the Interaction of Agro-Tourism Integration and Tourism-Driven Poverty Reduction: Empirical Evidence for China

**Debin Ma** [1], **Dongqi Sun** [2,*] and **Ziyi Wang** [3]

1   School of Business, Nanjing Normal University, Nanjing 200234, China; 220501005@njnu.edu.cn
2   Institute of Geographic Sciences and Natural Resources Research, Chinese Academy of Sciences, Beijing 100101, China
3   School of Architecture and Design, China University of Mining and Technology, Xuzhou 221116, China; wangziyi1011@smail.nju.edu.cn
*   Correspondence: sundq@igsnrr.ac.cn; Tel.:+86-186-1272-9027

**Abstract:** Under the robust impetus of China's rural revitalization strategy, agro-tourism integration and tourism-driven poverty reduction have profoundly impacted various aspects of China's economy, society, and ecology. This has propelled coordinated urban–rural development and the sustainable development of the tourism industry. This study introduces an analytical framework encompassing tourism-driven poverty reduction, agro-tourism integration, and rural revitalization. Through PVAR and threshold models, it empirically examines the interactive effects, dynamic relationships, and threshold effects between agro-tourism integration, tourism-driven poverty reduction, and rural revitalization. The conclusions are as follows: Firstly, the indices of rural revitalization and the level of agro-tourism integration show an increasing trend across Chinese provinces, with varying trends in tourism-driven poverty reduction efficiency. Secondly, there is a significant dynamic relationship among these factors, with the explanatory power of tourism-driven poverty reduction and agro-tourism integration gradually strengthening. Agro-tourism integration is identified as the most effective means of driving rural revitalization, while tourism-driven poverty reduction has a relatively weaker direct impact. Thirdly, tourism-driven poverty reduction exhibits a positive impulse response to agro-tourism integration. The improvement in tourism-driven poverty reduction efficiency propels further development in agro-tourism integration, thereby fostering rural revitalization. The efficiency of tourism-driven poverty reduction presents a single threshold effect in the process of agro-tourism integration promoting rural revitalization. Fourthly, the development of China's tourism industry has become an indispensable means of promoting rural revitalization and poverty reduction. However, rural revitalization is a comprehensive project influenced by various factors, requiring improvements and development across multiple aspects.

**Keywords:** agro-tourism integration; tourism-driven poverty reduction; rural revitalization; PVAR model; threshold model

## 1. Introduction

As the world experiences accelerated industrialization and urbanization, many countries are grappling with common issues in their rural areas. These issues include lagging rural governance, a monotonous development of industrial structures, lower living standards for rural residents, and significant ecological pollution in rural environments [1]. To achieve a more balanced, sustainable, and inclusive rural development, authorities and organizations worldwide have implemented a series of measures to promote rural development and drive rural revitalization [2]. For instance, the Food and Agriculture Organization of the United Nations (FAO) provides crucial support for rural development

in various countries through policy assistance and technological cooperation. The International Fund for Agricultural Development (IFAD) offers loans and technical support to improve livelihoods and agricultural production in rural communities. The European Union has identified sustainable agriculture and rural development as one of its regional sustainable development goals. The United Nations' Sustainable Development Goals (SDGs) focus on objectives such as poverty reduction, employment generation, and food security, leading global efforts to address rural revitalization and rural development. These international organizations and initiatives collectively underscore the global community's strong commitment to rural revitalization and emphasize the pivotal role of rural development in achieving sustainable development goals. Today, rural revitalization stands as a crucial strategic direction in China's current development agenda. In alignment with this overarching strategic goal, the Chinese government has undertaken a series of top-level designs and strategic plans. These include the "Central Document No. 1 of 2018" and the "National Rural Revitalization Strategy Plan (2018–2022)," both of which provide visionary policy blueprints. They unambiguously state that "prosperous industries" and "improved quality of life" are two essential facets of the comprehensive requirements for rural revitalization [3]. The basis of rural revitalization is the prosperity of industry, the core of which lies in how to stimulate the economic vitality of rural areas and use the various resources in rural areas to transform them into productive forces, which is of great importance to promote rural development and maintain the stability of rural society [4]. With the continual development of China's rural revitalization construction, the tourism industry is playing an increasingly prominent role in promoting the economic, cultural, and social progress of China's rural areas, among which, agro-tourism integration and tourism-driven poverty reduction are two important measures to promote the development of rural revitalization [2]. As a new mode of tourism development, agro-tourism integration has received wide attention and discussion. For most rural areas, agro-tourism integration is becoming an effective means to realize the transformation and upgrading of the rural economy, promote the development of rural industries, and solve the problems of agricultural development, providing a new way of development for the integration of three industries in rural areas [5,6]. In addition, tourism, as an important pillar industry for rural revitalization, is not only an important part of national economic development but also an important way to promote poverty eradication [7]. Tourism can drive the development of other industries in rural areas, promote farmers' employment and entrepreneurship, increase their income and become rich, and then help rural areas eradicate poverty [8]. The rapid development of tourism-driven poverty reduction in China has contributed to the reduction in poverty in China and has even made a great contribution to the fight against poverty worldwide [9].

Nowadays, the development of China's rural tourism industry is gaining momentum. In the context of rural revitalization and industrial integration, how to further realize the goal of promoting tourism with agriculture, developing agriculture with tourism, and even enriching agriculture with tourism through the integration of agriculture and tourism, thus promoting rural revitalization, is an important issue that China needs to pay attention to at present. Therefore, it is necessary to demonstrate whether agro-tourism integration and tourism-driven poverty reduction can drive rural revitalization and to elucidate the nature of the relationships among these three elements. From a practical standpoint, in-depth analysis of the mechanisms by which tourism-driven poverty reduction and agro-tourism integration contribute to rural revitalization, empirical investigation of current issues, and the proposal of scientifically sound policy recommendations are significant. These efforts guide the rational development of rural tourism across China's regions, expedite the country's modernization, and advance rural revitalization. Simultaneously, the findings can serve as experiential references for the development of related sectors in other nations worldwide. From a theoretical perspective, the development of tourism-driven poverty reduction and agro-tourism integration underscores the principle of social responsibility, highlights sustainable development goals, and encourages the tourism

industry to shoulder the mission of social development while propelling economic growth. This solidifies the foundation for social sustainability, further enriching the theory of sustainable tourism development.

The rest of this paper is organized as follows: Section 2 briefly reviews the research progress in rural revitalization, tourism-driven poverty reduction, and agro-tourism integration, and points out the gaps in the current research and the directions that need to be added. Section 3 is the analysis of the mechanism of action and research hypothesis. Section 4 is a presentation of the methodology and data sources. In Section 5, we report the results of the empirical analysis. Section 6 provides conclusions and discussion containing development recommendations.

## 2. Literature Review

Since the 1950s, Western countries have been focusing on the development and revitalization of rural areas and have implemented various rural revival policies [10]. These include the rural development policy implemented by the European Union, the rural agricultural development of the United Kingdom, the rural revitalization plan of France, and the new village movement of South Korea, among others [11,12]. These experiences offer valuable insights for the formulation of China's rural revitalization policies. Regarding the study of rural revitalization, scholars have conducted a lot of research on the development significance, theoretical ideas, scientific connotation, development mode, implementation path, and influencing factors of rural revitalization from many perspectives [13–15], such as policy interpretation, regional development, urban–rural integration, and practical experience, which has greatly promoted people's goal recognition and theoretical cognition of rural revitalization [16–18]. As a major national development strategy, China's rural revitalization emphasizes the question of "how to better develop the countryside" [19]. Even though many Chinese scholars have studied the connotation and evaluation of rural revitalization from international and domestic perspectives, how to better promote rural revitalization is still a key issue that needs to be discussed in depth [20].

Tourism-driven poverty reduction is a special way of alleviating poverty by developing tourism to drive economically underdeveloped areas out of poverty [21]. Research related to tourism-driven poverty reduction began with Peters [22] and De Kadt [23], and then the issue of tourism-driven poverty reduction began to enter the research horizon of scholars, whose research mainly focused on the meaning of tourism-driven poverty reduction, the basic theory, and the feasibility of development [24,25]. As the research on poverty reduction in tourism continues to deepen, the research results are becoming richer and richer, its theoretical aspects are gradually maturing, and a systematic theoretical framework and mechanism model have been formed [2]. As an important way to alleviate poverty in China, tourism has become a consensus to promote rural revitalization in poor areas [26,27]. However, how to improve the efficiency of tourism-driven poverty reduction on the premise of ensuring the stability of tourism-driven poverty reduction and consolidating the results of poverty reduction in order to give full play to its role in the comprehensive promotion of rural revitalization strategies has been an important focus of academic attention in recent years. Regarding the efficiency of tourism-driven poverty reduction, scholars mainly carry out research in terms of measuring objects, measuring methods, and influencing factors [28–30]. In general, the object of tourism-driven poverty reduction efficiency measurement changes from macro to micro, and the data envelopment method is the mainstream method of tourism-driven poverty reduction efficiency measurement [31,32]. The influencing factors cannot be unified by different stakeholders and micro and macro environments [33,34].

Realizing the integration of rural-related industries is an important path to the economic development of rural areas. Promoting the integration of agriculture and tourism is an important breakthrough to comprehensively promote rural revitalization in the future [21]. The concept of agro-tourism integration originated in the "civic paradise" in Germany in the mid-19th century. Since then, with the development of agritourism, the

study of agro-tourism integration has gradually emerged, and Hermans has studied the necessity of agro-tourism integration and pointed out that it is an organic integration between agriculture and tourism [35]. Waever and Fennell also pointed out that agro-tourism integration is a new type of industry that absorbs the characteristics of both agriculture and tourism [36]. By combing through the relevant literature, it can be seen that there are abundant theoretical and empirical research results on agro-tourism integration in academia, mainly focusing on the mode, path, mechanism, level, and influencing factors of mutual integration between the two industries [37–39]. At the same time, many studies have shown that the development of agro-tourism integration plays a significant role in promoting the development of rural areas. Fleischer and Tchetchik argue that agro-tourism integration can drive the modernization of agriculture by promoting the development of tourism, diversifying agriculture–tourism products to meet the diverse needs of agriculture–tourism tourists, expanding consumption, and thus stimulating the rapid development of the local economy [40]. Zhong and Tang believe that agro-tourism integration can promote the optimization and upgrading of rural industrial structures [41]. However, as industrial integration will encounter a series of dilemmas and problems in the process of concrete implementation, agro-tourism integration in the field of practice still has a long way to go.

In summary, numerous scholars have conducted a lot of research around the themes of tourism-driven poverty reduction, agro-tourism integration, and rural revitalization, but the following shortcomings still exist: Firstly, most of the existing studies focus on the relationship between tourism-driven poverty reduction and rural revitalization and agro-tourism integration and rural revitalization, but there is less literature that places the three in the same framework for comprehensive analysis. Secondly, the existing literature mostly focuses on the analysis of the connotation and measurement of the three but lacks quantitative analysis and empirical evidence on their relationship, and the research perspective needs to be expanded. Finally, although the concepts of tourism-driven poverty reduction and agro-tourism integration have been proposed long ago, in the practice of promoting rural revitalization, there are still problems such as the unclear relationship between them, a lack of holistic planning and unified guidance, and constraints by objective factors such as different regions, cultures, and natural conditions. Given this, this paper takes the 31 provinces and cities in China as the research objects and uses the entropy method, PVAR model, and threshold model to analyze whether tourism-driven poverty reduction and agro-tourism integration can promote rural revitalization, and the interrelationships among them. The aim is to enrich the research perspectives and theoretical frameworks in relevant areas, deepen our understanding of the relationships among these elements, and provide Chinese experience for the development of rural tourism and rural revitalization in other countries.

## 3. Mechanism of Action and Research Hypothesis

Numerous studies have shown that the development of rural tourism is an important way to promote rural economic development [40]. The development of tourism can drive the growth of the local economy and create more employment opportunities, thus improving the living standards of residents and achieving the purpose of poverty alleviation and reduction [42,43]. So, tourism-driven poverty reduction is one of the important factors in promoting rural revitalization. In general, tourism-driven poverty reduction is mostly supported by government policies [29]. Government policies on finance, taxation, and land can improve the soft environment for rural tourism development and provide convenient conditions for tourism development. The rise of tourism brings numerous consumptions and investments. Among them, the diversification of agricultural products and tourism products can improve the economic income of local residents, while the introduction of foreign investment can improve the tourism infrastructure in rural areas while enhancing the quality of local tourism services and the international attractiveness and competitiveness of rural tourism, thus promoting rural revitalization [38]. At the same time, the development of rural revitalization is also conducive to the promotion of tourism and poverty reduction.

In the context of rural revitalization, rural areas can integrate various local resources to provide more diversified and distinctive tourism resources and services for the tourism industry, attract more tourists to tourism consumption, drive the development of local agriculture, the handicraft industry, cultural creativity, and other industries, develop the sales market for local special agricultural products and handicrafts, improve economic and social benefits, and realise tourism-driven poverty reduction [44]. Based on these, this paper proposes hypotheses 1 and 2:

**Hypothesis 1:** *The better the development of tourism-driven poverty reduction, the more it can promote the development of rural revitalization.*

**Hypothesis 2:** *The better the development of rural revitalization, the more it will promote the development of tourism-driven poverty reduction.*

Agro-tourism integration is also an important way to develop rural tourism, and through the mutual combination of the tourism and agriculture industries, it can realize resource sharing, complementary advantages, and win–win benefits, which are of great significance to the development of rural revitalization [37,41]. Firstly, under the industry-driven effect, tourism can drive the development of local agriculture and service industries, creating more employment opportunities and increasing farmers' income, while agriculture can also provide tourism with high-quality ingredients and raw materials, improving the quality of tourism products and services. Secondly, under the effect of industrial integration, tourism can incorporate agricultural resources and agricultural culture into tourism products and services to improve the cultural connotation and experience value of tourism. In turn, agriculture can introduce the service concept and management experience of tourism into agricultural production and operation to improve the modernization level and market competitiveness of agricultural production [45]. Finally, under the effect of industrial upgrading, agriculture and tourism integration can improve the added value of agricultural products and drive agricultural production in the direction of high quality and high added value, while the integration of agriculture and tourism can attract more investment and talent into rural areas, further promote the upgrading of rural industrial structure, and promote rural economic development. At the same time, as an important strategy for promoting rural development in China, relevant government policies and plans provide important directions for the development of agriculture and tourism, while the government's promotion of rural tourism and agricultural products has increased the popularity and reputation of rural tourism, providing a broader development space for tourism and agriculture, and further promoting the rapid development of agricultural tourism integration [46,47]. Based on these, this paper proposes hypotheses 3 and 4:

**Hypothesis 3:** *The better the integration of agriculture and tourism, the more it can promote the development of rural revitalization.*

**Hypothesis 4:** *The better the development of rural revitalization, the more it can promote the integration of agriculture and tourism.*

Tourism as a medium has an impact on both tourism-driven poverty reduction and agro-tourism integration [48,49]. On the one hand, the efficient implementation of tourism-driven poverty reduction can effectively promote the integration of agriculture and tourism. The improvement in tourism-driven poverty reduction efficiency means that farmers' income will increase, and the increase in income will drive the upgrading of rural consumption. From the perspective of tourism, the diversified demand for consumption can force the upgrading of the structure of the industry and promote the development of agriculture and tourism integration. In addition, tourism-driven poverty reduction can eliminate absolute poverty in rural areas, provide a good development environment for industrial development, promote the docking of tourism with agriculture, animal husbandry, and

other related industries, form industrial interaction and linkage effects, and further promote the integration of agriculture and tourism. On the other hand, the integration of agriculture and tourism can promote the transformation and upgrading of the industry through the organic combination of agriculture and tourism, thus improving economic efficiency and achieving the purpose of eradicating poverty [8,50]. Agriculture and tourism are naturally complementary industries, with tourism being backwardly related to catering, accommodation, and handicrafts and forwardly related to agriculture. Tourism can extend the industrial chain through the mechanism of interaction between various industries, promote the optimization and upgrading of industrial structure, transformation, and adjustment, and bring greater economic benefits [51]. Based on these, this paper proposes hypotheses 5 and 6:

**Hypothesis 5:** *The better the development of tourism-driven poverty reduction, the more it will promote the development of agro-tourism integration.*

**Hypothesis 6:** *The better the development of agro-tourism integration, the more it will promote the development of tourism-driven poverty reduction.*

In summary, tourism-driven poverty reduction, agro-tourism integration, and rural revitalization development are closely linked, and there is an important practical basis for analyzing the three in the same framework (Figure 1).

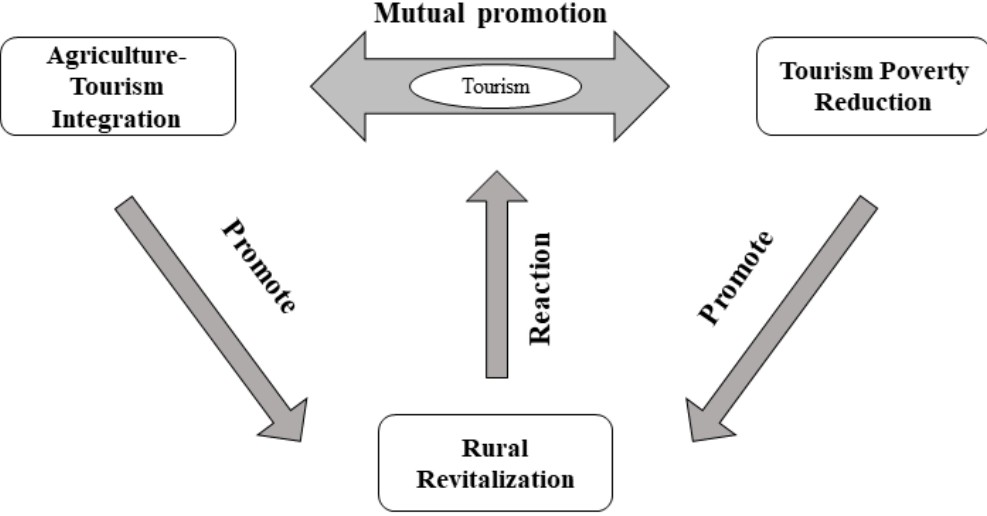

**Figure 1.** Mechanism of agro-tourism integration, tourism-driven poverty reduction, and rural revitalization.

## 4. Data and Methods

### 4.1. Research Methods

4.1.1. Entropy Value Method

Dimensionless Calculation

$$x'_{ij} = \frac{x_{ij} - \min(x_{1j}, x_{2j}, \ldots, x_{nj})}{\max(x_{1j}, x_{2j}, \ldots, x_{nj}) - \min(x_{1j}, x_{2j}, \ldots, x_{nj})} \tag{1}$$

$$x'_{ij} = \frac{\max(x_{1j}, x_{2j}, \ldots, x_{nj}) - x_{ij}}{\max(x_{1j}, x_{2j}, \ldots, x_{nj}) - \min(x_{1j}, x_{2j}, \ldots, x_{nj})} \tag{2}$$

Weight Calculation

In this paper, the weights of the index system are calculated using the entropy value method [52], assuming that there are *i* programs with *j* indicators designed, and $X_{ij}$ denotes

the $j$-th indicator value of the $i$-th program. The calculation steps of the entropy value method are as follows:

① Using the dimensionless processed data, the characteristic weight $p_{ij}$ is calculated for the $i$-th scenario under the $j$-th indicator.

$$p_{ij} = \frac{x_{ij}}{\Sigma_{i=1}^{n} x_{ij}} \; (i = 1, 2, \ldots, n; j = 1, 2, \ldots, m) \tag{3}$$

② Calculating the entropy value of the $j$-th indicator.

$$e_j = -k\Sigma_{i=1}^{n} p_{ij} ln(p_{ij}), \; k > 0, k = \frac{1}{1n(n)}, \; e_j \geq 0 \tag{4}$$

③ Calculating the coefficient of variability.

$$g_j = \frac{1 - e_i}{m - E_e}, \; E_e = \Sigma_{j=1}^{m} e_i, \; 0 \leq g_j \leq 1, \Sigma_{j=1}^{m} g_i = 1 \tag{5}$$

④ Calculating the weights of each evaluation index.

$$w_i = \frac{g_i}{\Sigma_{j=1}^{m} g_i} \; (1 \leq j \leq m) \tag{6}$$

⑤ Calculate the comprehensive score:

$$s_j = \Sigma_{j=1}^{m} w_j \times p_{ij} \; (i = 1, 2, \ldots, n) \tag{7}$$

4.1.2. Coupled Coordination Model

In this paper, we analyze the coupling and coordination of agro-tourism integration by establishing several indicators. With reference to relevant concepts and coefficient models [53], the coupling degree function of agro-tourism integration is established as follows:

$$C = \sqrt{\mu_1 \times \mu_2} / (\mu_1 + \mu_2) \tag{8}$$

where $\mu_1$ denotes the comprehensive score value of agriculture, $\mu_2$ denotes the comprehensive score value of tourism, and C denotes the coupling degree of agro-tourism integration, where $0 \leq C \leq 1$.

To better reflect the relationship between agriculture and tourism, the coupled coordination function is introduced:

$$T = \alpha\mu_1 + \beta\mu_2 \tag{9}$$

where $\alpha$ and $\beta$ are coefficients to be determined, assuming that agriculture and tourism are of equal importance and $\alpha = \beta = 1/2$.

$$D = \sqrt{C \times T} \tag{10}$$

According to the value of the coupling coordination degree, the coupling coordination degree can be roughly divided into six stages, as shown in Table 1.

**Table 1.** Division of coupled coscheduling.

| Coordination | Coordination Level | Coordination | Coordination Level |
|---|---|---|---|
| $0.0 < D \leq 0.2$ | Severe disorder | $0.4 < D \leq 0.6$ | Primary coordination |
| $0.2 < D \leq 0.3$ | Mild disorder | $0.6 < D \leq 0.8$ | Intermediate coordination |
| $0.3 < D \leq 0.4$ | Barely coordinated | $0.8 < D \leq 1.0$ | Senior coordination |

### 4.1.3. Super-Efficient SBM Model

In this paper, we study the efficiency of tourism-driven poverty reduction, expecting output maximization, so we adopt the output-oriented Super-SBM model based on the output-oriented SBM. Compared with the traditional DEA model, the Super-SBM model not only adds slack variables to the objective function, but the measured efficiency values can also break the 0–1 interval blocking, thus making the data more comparable. To this end, a super-efficient SBM model including non-desired outputs is used to measure eco-efficiency, and the specific steps can be found in the literature [54].

### 4.2. Indicator System Construction

#### 4.2.1. Construction of Rural Revitalization Index System

With comprehensive reference to the relevant research results [55], combined with the new trend of current rural revitalization research, and following the principles of scientificity and accessibility of data, this study constructs a rural revitalization evaluation index system based on five dimensions: flourishing industry, ecological livability, rural civilization, effective governance, and affluent living (Table 2).

**Table 2.** Indicator system for rural revitalization.

| | Primary Indicators | Secondary Indicators | Unit |
|---|---|---|---|
| Rural Revitalization | Flourishing industry | Total power of agricultural machinery | Million Kilowatts |
| | | Agriculture, forestry, and fishery production value | 100 million yuan |
| | | Effective irrigated area | Hectares |
| | | Grain production | 10,000 tons |
| | Ecological livability | Rural sanitary toilet penetration rate | % |
| | | Amount of pesticides used | Ton |
| | | Total gas production from biogas digester | 10,000 cubic meters |
| | | Total solar water heater area | 10,000 cubic meters |
| | Rural civilization | Number of cultural stations in townships | Pcs |
| | | Number of village health offices | Pcs |
| | | Number of rural elderly service institutions | Pcs |
| | | Illiteracy rate | % |
| | Effective governance | Ratio of per capita disposable income of urban and rural residents | % |
| | | General public budget expenditure | 100 million yuan |
| | | Rural minimum living security expenditure | Ten thousand yuan |
| | | Number of village committees | Pcs |
| | Affluent living | Rural disposable income per capita | Yuan |
| | | Rural Engel coefficient | % |
| | | Rural electricity consumption | Twh |
| | | Per capita consumption expenditure of rural residents | Yuan |

#### 4.2.2. Agro-Tourism Integration Index System Construction

Regarding the selection of indicators for measuring the development level of agro-tourism integration, this study draws on existing research ideas and methods and believes that the essence of agro-tourism integration development is the integration of agriculture and tourism, the process of cross-fertilization of industries, and that agro-tourism integration is not only conducive to solving the bottleneck problems of agriculture and tourism development but can also optimize the industrial structure and change the industrial growth mode, promote rural economic development, and respond to the rural revitalization strategy [56]. Based on this, this study constructs an evaluation index sys-

tem of agro-tourism integration from both agriculture and tourism, which contains five primary indicators of agricultural production, rural life, environment and ecology, tourism industry performance level, and tourism industry element level, and a total of 20 secondary indicators (Table 3).

**Table 3.** Index system for agriculture and tourism integration.

| Subsystems | Primary Indicators | Secondary Indicators | Unit |
|---|---|---|---|
| | Agricultural production | Total output value of primary industry | Ten thousand yuan |
| | | Grain yield | 10,000 tons |
| | | Arable land area | Hectares |
| | | Total power of agricultural machinery | Million Kilowatts |
| Agriculture | Rural life | Rural Engel coefficient | % |
| | | Rural disposable income per capita | Yuan |
| | | Per capita consumption expenditure of rural residents | Yuan |
| | | Rural minimum living security expenditure | Yuan |
| Agriculture and tourism integration | Environment and ecology | Fertilizer application amount | 10,000 tons |
| | | Rural electricity consumption | Twh |
| | | Rural sanitary toilet penetration rate | % |
| | | Effective irrigated area | Hectares |
| | Tourism industry performance level | Total tourism revenue | Ten thousand yuan |
| | | International tourism foreign exchange earnings | Millions of dollars |
| Tourism | | Total number of tourists | 10,000 person times |
| | | Number of inbound tourists received by foreigners | 10,000 person times |
| | Tourism industry element level | Number of travel agencies | Pcs |
| | | Number of star-rated hotels | Pcs |
| | | Number of employees in star-rated hotels | Per person |
| | | Number of travel agency employees | Per person |

4.2.3. Tourism-Driven Poverty Reduction Indicator System Construction

In this paper, we construct the tourism-driven poverty reduction index system from both input and output aspects [57], among which, the input indicators are tourism income per capita and the number of tourists received per capita, which mainly reflect the development status of local tourism and can reflect the development potential of tourism; the output indicators reflect the degree of impact of tourism on the local economy and on the economic income, medical care, culture, and quality of life of community residents, including the disposable income per capita of rural residents, urbanization rate, GDP per capita, the average number of students in colleges and universities per 100,000 people, and the number of beds in medical institutions (Table 4).

**Table 4.** Tourism-driven poverty reduction indicator system.

| | Indicator Types | Evaluation Indicators | Unit |
|---|---|---|---|
| Tourism-driven poverty reduction | Input indicators | Per capita tourism income | Yuan |
| | | Per capita tourist reception | Per person |
| | Output indicators | The disposable income per capita of rural residents | Yuan |
| | | Urbanization rate | % |
| | | GDP per capita | Yuan |
| | | The average number of students in colleges and universities | Per 100,000 people |
| | | The number of beds in medical institutions | Million |

*4.3. Model Setting*

4.3.1. Panel Data Vector Auto Regression Model

The PVAR model has become a common econometric tool in macroeconomic analysis. Unlike the traditional VAR model, the PVAR model takes into account both individual heterogeneity and individual time effects and is able to provide a better description of the correlation between variables [58]. The specific form is as follows:

$$Y_{it} = \sum_{l=1}^{P} \Phi_l Y_{i,t-l} + \gamma_i + u_{it} \tag{11}$$

where $Y_{it}$ is the explained variable, $P$ denotes the lag order, $\Phi_l$ denotes the slope coefficient of lag l period, $\gamma_i$ is the individual fixed effect, and $u_{it}$ is the random error term.

4.3.2. Threshold Model

The "threshold effect" is based on the threshold estimates and examines the effect of each variable on the explanatory variables in the range of different zone systems [59]. Its advantage is that the threshold regression models constructed based on threshold variables are more accurate and scientific in fitting the nonlinear relationships among the variables due to the threshold effects. After understanding the basic relationship between the three, the control variables were further introduced to investigate the nonlinear relationship and the threshold model shown below was constructed:

$$rev_{i,t} = \alpha_i + \beta_1 tpr_{i,t} \cdot I(mean > \gamma) + \beta_2 tpr_{i,t} \cdot I(mean < \gamma) + \beta_3 ati_{i,t} + \beta_4 eco_{i,t} + \beta_5 urb_{i,t} + \beta_6 ind_{i,t} + \beta_7 road_{i,t} + \beta_8 gov_{i,t} + \beta_9 edu_{i,t} + \beta_{10} as_{i,t} + \beta_{11} fdi_{i,t} + \mu_{i,t} \tag{12}$$

where $i$ denotes the province, $t$ denotes the year, $\gamma$ denotes an unknown threshold, $I(\cdot)$ denotes the indicator function, $\mu_{i,t}$ is a random perturbation term, and other variables are defined as shown in Table 5.

**Table 5.** Threshold model variables.

| Variables | Symbol | Name of Variables | Definition |
|---|---|---|---|
| Explained variable | Rev | Rural revitalization | |
| Explanatory variable | Tpr | Tourism-driven poverty reduction | Calculated by the above method |
| | Ati | Agriculture and tourism integration | |
| Control variable | Eco | Economic development level | GDP growth rate |
| | Urb | Urbanization rate | Percentage of urban household population in total population |
| | Ind | Industry structure | Percentage of employment in tertiary sector to total employment |
| | Road | Road density | Ratio of road mileage to land area |
| | Gov | Level of financial support for agriculture | The proportion of financial agricultural expenditure to financial expenditure |
| | Edu | Human capital stock | Average years of education |
| | As | Agricultural structure | The ratio of sown area of grain to total sown area of crops |
| | Fdi | Level of regional opening to the outside world | Percentage of total imports and exports to GDP |

### 4.4. Data Sources and Descriptive Statistics

The data of 31 provinces of China for 15 years from 2006 to 2020 are selected as the research objects (excluding Hong Kong, Taiwan, and Macao), and all data are obtained from the China Urban Statistical Yearbook, China Environmental Statistical Yearbook, China Rural Statistical Yearbook, China Health and Health Statistical Yearbook, Regional Economic Statistical Yearbook, China Urban Construction Statistical Yearbook, and the few missing values are supplemented by the interpolation method. The definition and descriptive statistics of each variable are shown in Table 6.

**Table 6.** Descriptive statistics.

| Name of Variables | Sample Size | Mean Value | Standard Error | Min. Value | Max. Value |
|---|---|---|---|---|---|
| Rev | 450 | 0.228 | 0.123 | 0.0461 | 0.563 |
| Tpr | 450 | 0.278 | 0.286 | 0.00609 | 3.060 |
| Ati | 450 | 0.342 | 0.0788 | 0.145 | 0.510 |
| Eco | 450 | 9.732 | 3.614 | 0.200 | 19.20 |
| Urb | 450 | 54.40 | 14.54 | 21.13 | 89.60 |
| Ind | 450 | 39.81 | 11.01 | 14.93 | 83.10 |
| Road | 450 | 85.10 | 50.70 | 3.753 | 219.4 |
| Gov | 450 | 10.74 | 3.809 | 1.575 | 20.38 |
| Edu | 450 | 8.810 | 1.235 | 4.161 | 12.78 |
| As | 450 | 65.49 | 13.19 | 32.81 | 97.08 |
| Fdi | 450 | 0.304 | 0.348 | 0.00764 | 1.712 |

## 5. Analysis of Empirical Results

### 5.1. Analysis of the Results of Rural Revitalization Index, Tourism-Driven Poverty Reduction Efficiency, and Agro-Tourism Integration Level

5.1.1. Analysis of Rural Revitalization Index Results

The entropy method formula is used to measure the rural revitalization index of 31 provinces in China from 2006 to 2020, but due to the large time span of the selected data, the data from 2006 to 2020 are divided into 2006–2010, 2011–2015, and 2016–2020. The average values of the rural revitalization indices of each province, each region, and the whole country in the three periods are calculated, respectively, to visualize the trends of the study area over the years (Figure 2 and Table 7). In terms of time, the rural revitalization index of all provinces in China shows a steady increase overall. It is noteworthy that the increase from the 2006–2010 phase to the 2011–2015 phase is greater than the increase from

the 2011–2015 phase to the 2016–2020 phase. In terms of space, there are obvious gaps in the rural revitalization index of each province, and the spatial distribution pattern of the central region > the eastern region > the northeastern region > the western region is shown. In addition, the rural revitalization index of most provinces in the eastern and central regions is significantly higher than the national average, while most provinces in the northeastern and western regions are significantly lower than the national average.

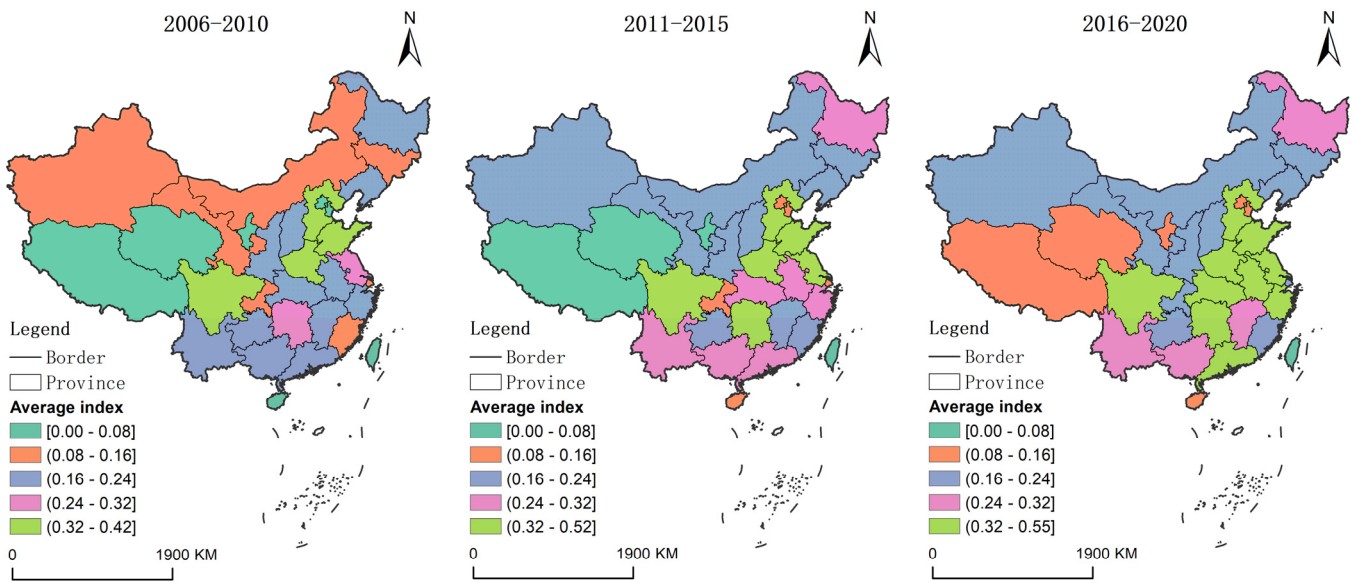

**Figure 2.** National average and regional rural revitalization index by stages from 2006 to 2020. Note: Produced based on the standard map GS(2019)1822 of the Ministry of Natural Resources Standard Map Service website, with no modifications to the base map.

**Table 7.** Average rural revitalization index by region from 2006 to 2020.

| Regions | 2006–2010 | 2011–2015 | 2016–2020 |
|---|---|---|---|
| National | 0.182 | 0.237 | 0.269 |
| Eastern | 0.199 | 0.260 | 0.297 |
| Northeast | 0.150 | 0.206 | 0.235 |
| Central | 0.240 | 0.306 | 0.338 |
| Western | 0.146 | 0.192 | 0.220 |

5.1.2. Analysis of Tourism-Driven Poverty Reduction Efficiency Results

According to the formula of the super-efficient SBM model, MaxDEA is applied to measure the tourism-driven poverty reduction efficiency, and the tourism-driven poverty reduction efficiency values are obtained for each region from 2006 to 2020 (Figure 3, Table 8). From the perspective of time change, the overall trend of tourism-driven poverty reduction efficiency in each province is more complex. Among them, Hubei, Hunan, Guangxi, Guangdong, Hainan, Hebei, Chongqing, and other provinces show a "U"-shaped trend of decreasing and then increasing, while Anhui, Jiangxi, and Fujian show an inverted "U"-shaped trend of increasing and then decreasing. The trend of Tibet, Sichuan, Yunnan, and Guizhou shows an "N"-type characteristic of decreasing, rising, and then decreasing. Notably, the efficiency of tourism in reducing poverty in Inner Mongolia has increased significantly, reflecting the growing importance of tourism development to the region's poverty reduction efforts. Spatially, there are obvious differences in tourism-driven poverty reduction efficiency among provinces, showing a spatial pattern of central > western > eastern > northeastern regions, and the poverty reduction efficiency values of central Henan, Hunan, and Hubei provinces exceed 1 in some years, indicating that the tourism

factors in these regions are more reasonably allocated and the tourism industry shows benign development trends.

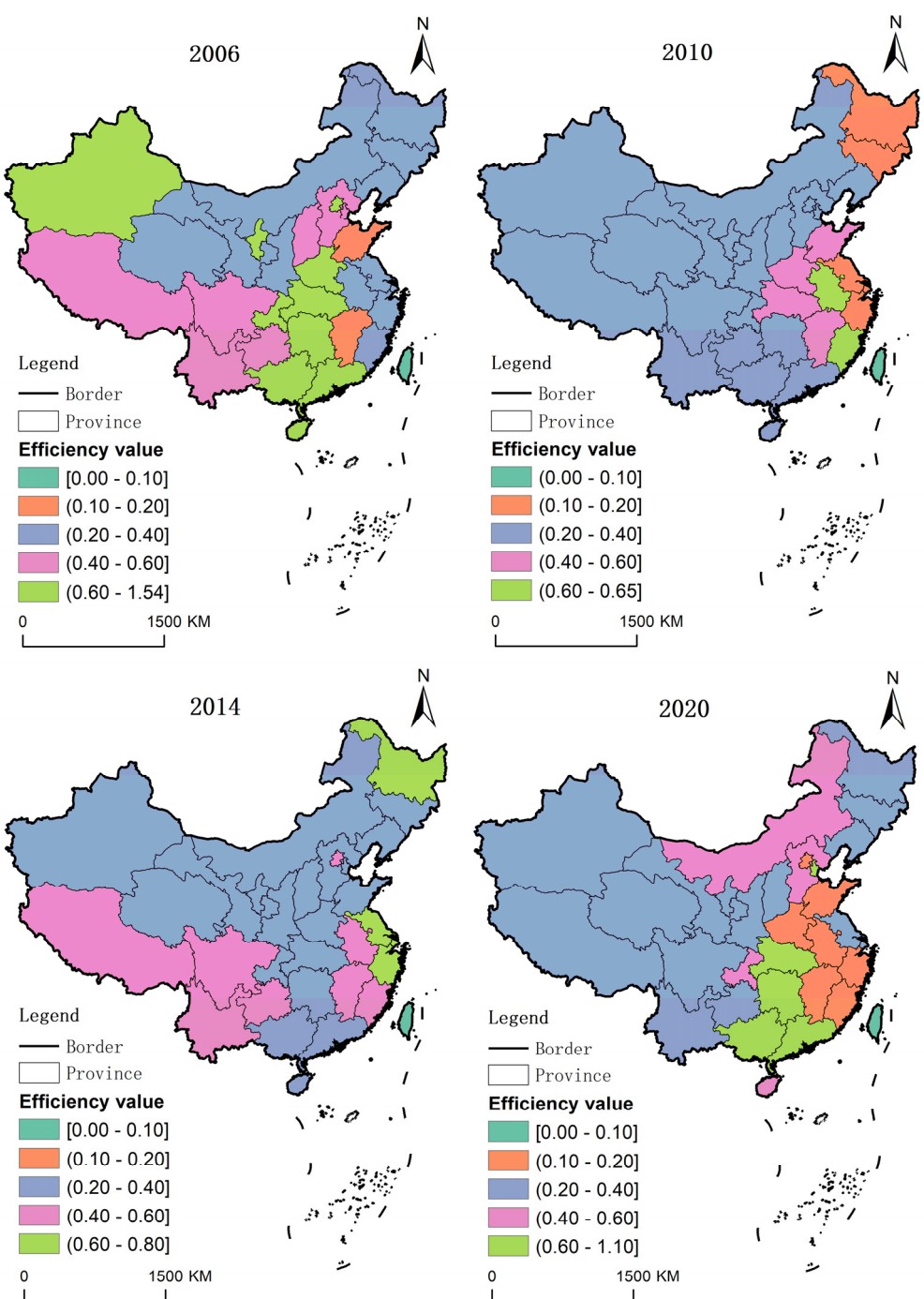

**Figure 3.** Tourism-driven poverty reduction efficiency values by region from 2006 to 2020. Note: Produced based on the standard map GS(2019)1822 of the Ministry of Natural Resources Standard Map Service website, with no modifications to the base map.

**Table 8.** Average tourism-driven poverty reduction efficiency values by region from 2006 to 2020.

| Regions | 2006 | 2010 | 2014 | 2020 | Mean |
|---|---|---|---|---|---|
| Eastern | 0.446 | 0.314 | 0.444 | 0.381 | 0.396 |
| Northeast | 0.319 | 0.191 | 0.483 | 0.308 | 0.325 |
| Central | 0.632 | 0.445 | 0.381 | 0.402 | 0.465 |
| Western | 0.649 | 0.296 | 0.371 | 0.346 | 0.415 |

5.1.3. Analysis of the Results of Agro-Tourism Integration

As can be seen from Table 9, the overall change in the level of agro-tourism integration in China from 2006 to 2020 shows a slow growth trend, moving from the disorder stage to the coordination stage, but the degree of agro-tourism integration in each stage shows obvious regional differences. From the results in 2020, the top three regions in the level of agro-tourism integration are Guangdong, Zhejiang, and Jiangsu, all of which are in the primary coordination stage, indicating that their agro-tourism integration development has a higher quality and better stability. Tibet, Qinghai, and Ningxia, however, have a lower level of agro-tourism integration and are still at a mild disorder stage. The reasons for this phenomenon may be as follows: The first is that China is a large country, and the geographical location and resource endowments of different regions differ greatly. The second is that the different levels of economic development in China's provinces have led to significant differences in rural per capita disposable income and total tourism income, resulting in uneven levels of synergistic development across regions.

**Table 9.** The results of agro-tourism integration by region from 2006 to 2020.

| Provinces | 2006 | 2010 | 2014 | 2020 | Provinces | 2006 | 2010 | 2014 | 2020 |
|---|---|---|---|---|---|---|---|---|---|
| Beijing | 0.334 | 0.357 | 0.379 | 0.395 | Hubei | 0.346 | 0.356 | 0.396 | 0.446 |
| Tianjin | 0.27 | 0.294 | 0.306 | 0.275 | Hainan | 0.338 | 0.353 | 0.412 | 0.455 |
| Hebei | 0.343 | 0.316 | 0.358 | 0.347 | Guangdong | 0.421 | 0.443 | 0.482 | 0.51 |
| Shanxi | 0.312 | 0.322 | 0.325 | 0.314 | Guangxi | 0.32 | 0.329 | 0.371 | 0.462 |
| Inner Mongolia | 0.293 | 0.311 | 0.332 | 0.351 | Hainan | 0.266 | 0.273 | 0.28 | 0.307 |
| Liaoning | 0.354 | 0.375 | 0.389 | 0.371 | Chongqing | 0.289 | 0.332 | 0.353 | 0.384 |
| Jilin | 0.275 | 0.289 | 0.319 | 0.298 | Sichuan | 0.346 | 0.341 | 0.413 | 0.459 |
| Heilongjiang | 0.296 | 0.308 | 0.324 | 0.29 | Guizhou | 0.257 | 0.283 | 0.352 | 0.434 |
| Shanghai | 0.351 | 0.366 | 0.4 | 0.424 | Yunnan | 0.36 | 0.359 | 0.385 | 0.425 |
| Jiangsu | 0.411 | 0.437 | 0.446 | 0.463 | Tibet | 0.177 | 0.187 | 0.198 | 0.243 |
| Zhejiang | 0.399 | 0.422 | 0.447 | 0.474 | Shaanxi | 0.314 | 0.332 | 0.357 | 0.385 |
| Anhui | 0.315 | 0.345 | 0.396 | 0.449 | Gansu | 0.266 | 0.272 | 0.275 | 0.286 |
| Fujian | 0.332 | 0.353 | 0.388 | 0.427 | Qinghai | 0.192 | 0.208 | 0.222 | 0.246 |
| Jiangxi | 0.306 | 0.326 | 0.386 | 0.434 | Ningxia | 0.145 | 0.15 | 0.196 | 0.218 |
| Shandong | 0.401 | 0.427 | 0.447 | 0.441 | Xinjiang | 0.29 | 0.296 | 0.308 | 0.287 |
| Henan | 0.162 | 0.178 | 0.193 | 0.219 | —— | | | | |

*5.2. Results and Analysis of PVAR Model*

Using the above model and variable data to empirically test the relationship between tourism-driven poverty reduction, agro-tourism integration, and rural revitalization in 31 provinces across China, the first-order difference of the core variables is required before conducting model regression, and a unit root test and panel cointegration test are conducted. The specific results are as follows:

5.2.1. Smoothing Test and Cointegration Test Analysis

In order to avoid "pseudo-regression" in the regression of time series, this paper uses the LLC test and the IPS test to conduct unit root tests for the core variables selected by the PVAR model. As can be seen from Table 10, the data for all variables passed the significance level test and had good smoothness. Then, the Kao cointegration test was performed on the variables to further identify the existence of a stable long-term relationship between

the variables. The results showed that all variables rejected the original hypothesis at the 1% significance level, indicating that there was a cointegration relationship between the variables, and that the PVAR model could continue to be constructed.

**Table 10.** Unit root test results of panel data.

| Variables | LLC Test | IPS Test | Smoothness |
|---|---|---|---|
| Rev | −6.3543 *** | −6.3359 *** | Smooth |
| Tpr | −2.4749 *** | −9.3859 *** | Smooth |
| Ati | −4.2416 *** | −9.1899 *** | Smooth |

Note: *** indicates significant at the 1% level.

### 5.2.2. The Determination of the Optimal Lag Order

As lag orders in the PVAR model have an important effect on each statistic, it is necessary to select the appropriate lag order before performing estimation. According to the existing research results (Table 11), the selection of the optimal lag order was based on the information minimization criterion [60]. In this paper, the Stata software (version 18) was used to calculate the corresponding Akuchi information criterion (AIC), Bayesian information criterion (BIC), and Hannan–Quinn information criterion (HQIC) for the case of model lag 1–3 periods, and the one with the smallest value of the three criteria should be selected as the optimal lag order of the model. From the results, it was shown that the optimal lag order is the first order.

**Table 11.** Final lag order selection results.

| Lag Order | AIC | BIC | HQIC |
|---|---|---|---|
| 1 | −6.15219 | −5.07765 * | −5.72546 * |
| 2 | −6.20314 * | −4.95581 | −5.70618 |
| 3 | −5.93989 | −4.49348 | −5.36168 |

Note: * indicates significant at the 10% level.

### 5.2.3. Granger Causality Test

To clarify the relationship between rural revitalization, tourism-driven poverty reduction, and agro-tourism integration, this study conducted a one-period lagged Granger causality test based on the constructed PVAR model. The results, presented in Table 12, revealed bidirectional Granger causality between rural revitalization and agro-tourism integration, indicating that they significantly impact each other. Therefore, rural revitalization can promote the improvement in agro-tourism integration, whilst agro-tourism integration can also improve the development of rural revitalization, confirming hypotheses 2 and 4. Additionally, rural revitalization was identified as the one-way Granger cause of tourism-driven poverty reduction, verifying hypothesis 2 but not hypothesis 1. This may be due to the unbalanced economic development in various regions of China, which makes the whole level of tourism-driven poverty reduction efficiency to be lower and the promotion of rural revitalization to be weaker. Furthermore, tourism-driven poverty reduction was found to be the one-way Granger cause of agro-tourism integration, substantiating hypothesis 5, while the reverse effect of agro-tourism integration on improving tourism-driven poverty reduction efficiency was not strong, and thus hypothesis 6 was not verified. Overall, the joint equations mostly passed the significance test, signifying significant interaction among rural revitalization, agro-tourism integration, and tourism-driven poverty reduction in each province of China and validating the model to some extent.

### 5.2.4. Analysis of the Impulse Response Results

Based on the above analysis, the impulse response function was used to further analyze the mutually dynamic relationship among the three. The impulse response function (IRF) refers to the degree of change in one endogenous variable when a unit pulse is applied to

another variable. It can also be understood as the impact of a change in an endogenous variable on the entire model. The IRF is used to analyze the dynamic transmission pathways between variables. It provides an intuitive depiction of the dynamic interactions between variables. Impulse response function plots for the lag 1 period were obtained by giving a 1 standard deviation shock to the variables using Monte Carlo simulations 200 times, and 95% confidence intervals are shown. In Figure 4, the horizontal axis represents the lag period, the vertical axis represents the degree of response of the endogenous variables, and the middle solid line represents the impulse response function. It was observed that the impulse response functions of all variables converged to zero after 10 periods, showing a convergence trend, which proved that the PVAR model was robust.

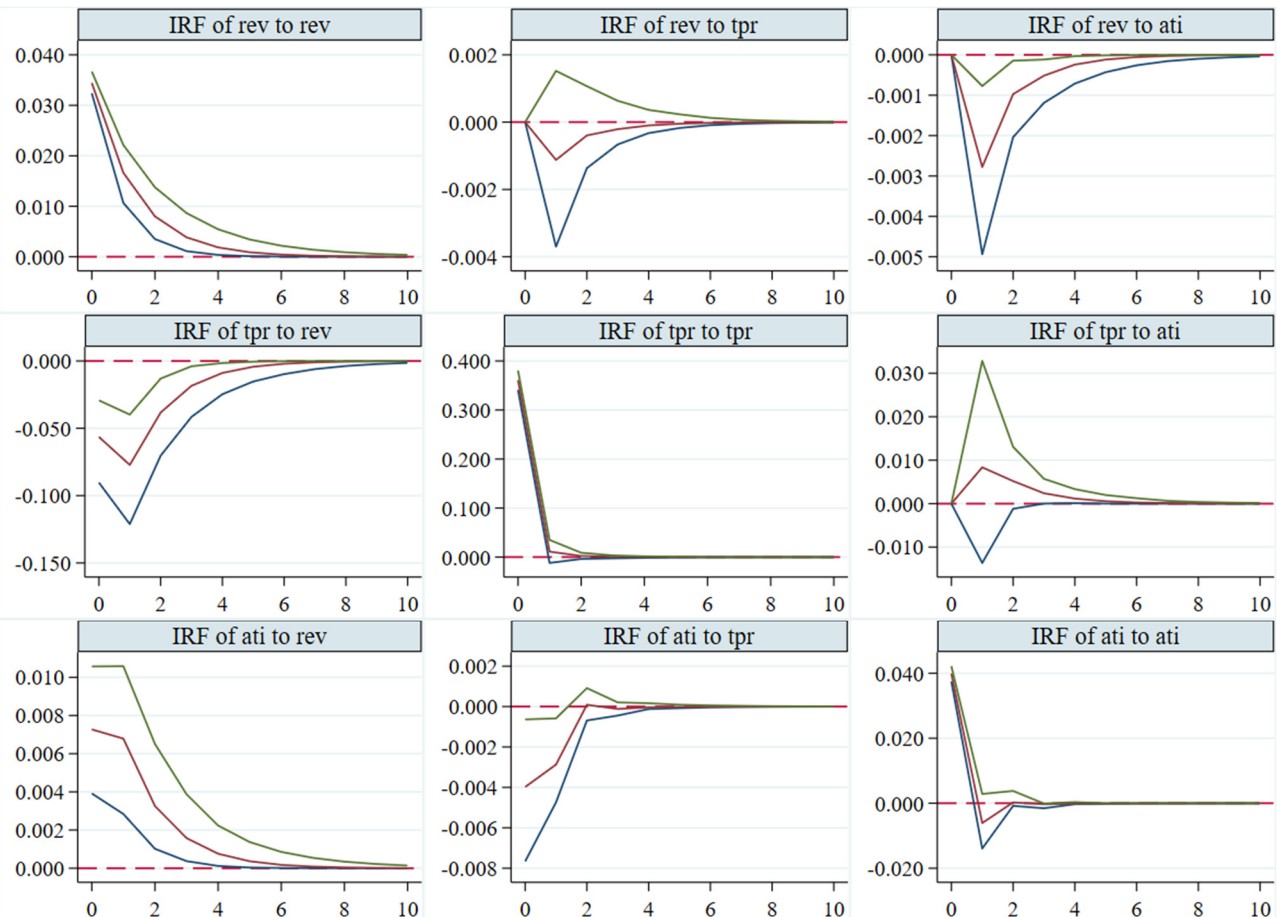

**Figure 4.** Pulse response of rural revitalization, tourism-driven poverty reduction, and integration of agriculture and tourism. Note: In the pulse response plot, the red line in the middle represents the estimation of the pulse response within the study time range, while the area between the green and blue lines is a standard error confidence band.

The responses of rural revitalization, agro-tourism integration, and tourism-driven poverty reduction to their own shocks showed that the response was rapid and positive, with a maximum value in the first period and then fluctuating down to zero value smoothly. It showed that rural revitalization, agro-tourism integration, and tourism-driven poverty reduction had a strong inertia for economic development and a good self-reinforcing effect.

**Table 12.** Granger causality test results.

| Variable | Original Hypothesis | Chi-Square | Conclusions |
| --- | --- | --- | --- |
| Rev | Tpr is not the cause | 0.956 | Accept |
| | Ati is not the cause | 4.689 ** | Reject |
| | All is not the cause | 4.972 * | Reject |
| Tpr | Rev is not the cause | 9.842 *** | Reject |
| | Ati is not the cause | 0.356 | Accept |
| | All is not the cause | 9.876 *** | Reject |
| Ati | Rev is not the cause | 9.040 *** | Reject |
| | Tpr is not the cause | 7.610 *** | Reject |
| | All is not the cause | 18.744 *** | Reject |

Note: * indicates significant at the 10% level, ** indicates significant at the 5% level, *** indicates significant at the 1% level.

According to the impulse response results between agro-tourism integration and rural revitalization, agro-tourism integration has a positive influence on rural revitalization, with the greatest influence at the beginning of the period, near 0.008, and gradually tending to zero after the fifth period, with the influence weakening over time. However, rural revitalization had a negative impact on the integration of agriculture and tourism, which reached its lowest point in the first period and steadily weakened in subsequent periods. This demonstrated that the promotion effect of agro-tourism integration on rural revitalization in China has not been fully implemented, as has the promotion effect between rural revitalization and poverty reduction in the tourism industry.

The results of the impulse response curves between tourism-driven poverty reduction and agriculture–tourism integration showed that tourism-driven poverty reduction had a positive effect on agriculture–tourism integration, but when agriculture–tourism integration was hit by tourism-driven poverty reduction, it showed a short-term negative response and then tended to zero, which was also consistent with the results of the Granger causality test, and to a certain extent, it reflected that tourism-driven poverty reduction could promote agriculture–tourism integration and then further promote rural revitalization.

5.2.5. Analysis of the Variance Decomposition Results

In panel data, different associations and influences can exist among individual entities or units. Therefore, we further conducted a variance decomposition of the PVAR model. This analysis yielded a coefficient matrix that encompasses relationships between various variables. Using these coefficients, we could assess the contributions of each variable to the variance, allowing us to examine the degree of mutual influence between tourism-driven poverty reduction, agro-tourism integration, and rural revitalization. The results are presented in Table 13. In general, the variance contribution rate of each variable depended mainly on its own development, with the contribution rates of tourism-driven poverty reduction and agro-tourism integration to rural revitalization increasing over time. The variance decomposition results for periods 5 and 6 were similar, indicating stabilization after period 5. Specifically, in period 5, rural revitalization contributed to itself at a 99.3% rate, while tourism-driven poverty reduction and agro-tourism integration contributed at rates of 7.8% and 6.4%, respectively. Rural revitalization's development was primarily self-influenced, with its own variance occupying a crucial position, while its explanatory power to tourism-driven poverty reduction and agro-tourism integration increased, suggesting the interaction between the three was strengthening.

**Table 13.** Variance decomposition results.

| Variables | Periods | D_Rev | D_Tpr | D_Ati |
|---|---|---|---|---|
| D_Rev | 1.000 | 1.000 | 0.000 | 0.000 |
| D_Tpr | 1.000 | 0.024 | 0.976 | 0.000 |
| D_Ati | 1.000 | 0.032 | 0.010 | 0.959 |
| D_Rev | 2.000 | 0.994 | 0.001 | 0.005 |
| D_Tpr | 2.000 | 0.065 | 0.934 | 0.000 |
| D_Ati | 2.000 | 0.056 | 0.014 | 0.930 |
| D_Rev | 3.000 | 0.993 | 0.001 | 0.006 |
| D_Tpr | 3.000 | 0.075 | 0.924 | 0.001 |
| D_Ati | 3.000 | 0.062 | 0.014 | 0.924 |
| D_Rev | 4.000 | 0.993 | 0.001 | 0.006 |
| D_Tpr | 4.000 | 0.077 | 0.922 | 0.001 |
| D_Ati | 4.000 | 0.063 | 0.014 | 0.923 |
| D_Rev | 5.000 | 0.993 | 0.001 | 0.006 |
| D_Tpr | 5.000 | 0.078 | 0.921 | 0.001 |
| D_Ati | 5.000 | 0.064 | 0.014 | 0.923 |
| D_Rev | 6.000 | 0.993 | 0.001 | 0.006 |
| D_Tpr | 6.000 | 0.078 | 0.921 | 0.001 |
| D_Ati | 6.000 | 0.064 | 0.014 | 0.923 |

*5.3. Analysis of the Threshold Effect Results*

The analysis of the results using the PVAR model indicates a positive relationship between tourism-driven poverty reduction, agro-tourism integration, and rural revitalization. The explanatory power of tourism-driven poverty reduction and agro-tourism integration on rural revitalization gradually increased. Agro-tourism integration and rural revitalization had bidirectional Greenland causality, while tourism-driven poverty reduction had a positive impulse response to agro-tourism integration. Improving the efficiency of tourism-driven poverty reduction could further promote the development of agro-tourism integration, which in turn would promote rural revitalization. However, due to significant differences in the efficiency of tourism-driven poverty reduction across China and imbalances in economic development and infrastructure, the intermediate factor of tourism-driven poverty reduction had varying effects on promoting rural revitalization through agro-tourism integration. Furthermore, rural revitalization may be influenced by multiple factors. This paper introduces control variables affecting rural revitalization development and analyzes the threshold values and influencing factors using the threshold regression model to determine the extent to which agriculture and tourism integration can effectively promote rural revitalization under the efficiency of tourism-driven poverty reduction.

5.3.1. Results of the Threshold Effect Test

We employ a threshold model for threshold effect testing. The threshold model is a useful tool for capturing nonlinear relationships in data. By identifying threshold values and related parameters, it can assist in explaining different behaviors and patterns in the data. This study investigated the threshold effect by estimating the model with single, double, and triple threshold assumptions. Tourism-driven poverty reduction efficiency served as the threshold variable, and 300 autosamples were conducted using STATA software. The results revealed a significant single threshold effect at the 1% level. As a result, the analysis utilized the single threshold model (Table 14).

**Table 14.** Threshold effect estimation results.

| Models | F Value | *p* Value | Threshold Value | | |
| --- | --- | --- | --- | --- | --- |
| | | | 10% | 5% | 1% |
| Single threshold | 90.44 | 0.0000 | 39.0001 | 63.2815 | 64.0229 |
| Double threshold | 21.22 | 0.2400 | 28.5180 | 40.3248 | 52.8463 |
| Triple threshold | 15.14 | 0.9000 | 55.9061 | 60.2995 | 61.0613 |

The estimates of the single threshold and the corresponding 95% confidence intervals are shown in Table 15. A single threshold value of $-1.0064$ for tourism-driven poverty reduction is obtained from the threshold estimation.

**Table 15.** Threshold estimation results.

| Threshold Value | Estimated Value | 95% Confidence Interval |
| --- | --- | --- |
| Single threshold | $-1.0064$ | $\{-1.0149, -0.9806\}$ |

5.3.2. Analysis of the Threshold Model Results

The parameter estimation results of the threshold model in Table 16 showed that the urbanization rate and the level of regional opening to the outside world had a significant negative inhibitory effect on rural revitalization. While the level of economic development, industrial structure, road density, level of financial support for agriculture, and human capital stock had significant positive effects on rural revitalization, agricultural structure had insignificant positive effects on rural revitalization. There was a single threshold in tourism-driven poverty reduction, and the significant contribution of agro-tourism integration to rural revitalization tended to increase step by step after reaching the threshold value. With tourism-driven poverty reduction below the threshold, the coefficient of $q_{it} < \gamma$ is 0.480, and agro-tourism integration plays a significant role in promoting rural revitalization. After exceeding the threshold, the coefficient of $q_{it} > \gamma$ grows to 0.570, and the role of agro-tourism integration in promoting rural revitalization is enhanced. The reason might be that with the rapid development of the rural economy, the optimization of rural industrial structure was promoted, the improvement in rural transportation facilities made the level of integrated development of agriculture and tourism further improved, the development of the agriculture and tourism industries provided more opportunities for labor force employment, alleviated poverty, improved the quality of life of farmers, and would create a strong attraction for talents, which in turn enhanced the capital stock, thus helping rural revitalization. Meanwhile, the increase in financial support for agriculture also provided sufficient financial guarantees for the development of rural revitalization. However, the influence of urbanization development on rural revitalization was weak, reflecting that there were still obstacles to the flow of economic development factors between urban and rural areas and that the efficiency of resource allocation still needed to be improved. In addition, the significantly negative level of external openness might be due to the poor foundation of the international environment for the opening up of China's countryside to the outside world, the insufficient foreign trade of rural agricultural products, and the need for further development of international tourism.

**Table 16.** Threshold model parameter estimation results.

| Explanatory Variables | Coefficient | t Value |
|---|---|---|
| Eco | 0.048 *** | −4.38 |
| Urb | −0.448 *** | −4.95 |
| Ind | 0.147 ** | 3.05 |
| Road | 0.410 *** | 6.77 |
| Gov | 0.111 *** | 7.38 |
| Edu | 0.783 *** | 6.00 |
| As | 0.092 | 1.23 |
| Fdi | −0.078 *** | −6.02 |
| qit < γ | 0.480 *** | 6.24 |
| qit > γ | 0.570 *** | 7.36 |
| C | −3.900 *** | −8.09 |
| N | 450 | 450 |

Note: ** indicates significant at the 5% level, *** indicates significant at the 1% level.

## 6. Conclusions and Discussion

### 6.1. Discussion

With the booming development of tourism in China, tourism has become an indispensable tool for promoting rural revitalization and poverty reduction. However, the direct contribution of tourism-driven poverty reduction to rural revitalization in China is still weak, probably due to the unbalanced development of China's regions, which makes tourism resources unevenly distributed, and the efficiency of tourism-driven poverty reduction varies widely. Although some regions are rich in tourism resources, the development of tourism is limited due to problems such as inconvenient transportation and poor infrastructure, making it difficult to fully play a role in poverty reduction. In addition, the traditional forms of tourism in most areas of China are still deeply rooted and lack innovation, transformation, and upgrading to meet the changing needs of the tourism market and the new consumer demands of tourists. Therefore, there is an urgent need to promote innovation, transformation, and upgrading of tourism in rural areas to improve the added value and competitiveness of tourism, and the integration of agriculture and tourism is the most effective way to do this. Through agro-tourism integration, we can effectively promote the transformation of rural economic structure, make the rural economy more diversified, promote the development of rural tourism and agriculture as well as the transformation of rural economic structure, increase the economic income of rural areas, improve the efficiency of tourism-driven poverty reduction, and in turn promote rural revitalization.

Furthermore, under the strong impetus of China's rural revitalization strategy, the integration of agriculture and tourism can provide a fresh perspective on the cultural value of agriculture. It highlights the contemporary significance and comprehensive benefits of agricultural culture, allowing it to blend with modern life from a tourist's viewpoint. Throughout the entire tourism process, agricultural culture is imprinted, making it more credible, tangible, and transmissible. The deep integration of agriculture and tourism in China allows advanced elements of the tourism industry to take root in various agricultural processes. Additionally, the tourism industry's reliance on ecological environments triggers a reassessment and amplification of the ecological value of agriculture. This contributes to enhancing agricultural ecological efficiency and opening new opportunities for China's agricultural modernization while promoting sustainable tourism development. However, rural revitalization is a systematic project that requires the joint efforts of all aspects of society, so China should also make efforts to promote rural revitalization from the following aspects: improving rural roads, railroads, and communications infrastructure, improving transportation, information, and other infrastructure; improving the rural financial system, expanding the coverage of rural financial services, and increasing the convenience of financing for farmers; integrating the coordinated development of urban and rural areas and promoting the optimal allocation of various resources; promoting the further opening of rural areas, strengthening scientific and technological innovation and technology

introduction, encouraging and supporting rural enterprises to go to the international market, and improving the international competitiveness of related products; strengthening human resources training, improving farmers' skills and quality through various means, vigorously cultivating new agricultural subjects, optimizing subsidy policies, and attracting college students and other highly qualified personnel to come to work and start businesses.

### 6.2. Conclusions

As two important means to promote rural revitalization, tourism-driven poverty reduction and agro-tourism integration are in line with the requirements of the United Nations' Sustainable Development Goals (SDGs). By promoting the development of tourism in rural areas, they provide local residents with ways to increase income and employment opportunities, effectively alleviate poverty, highlight the principle of social responsibility, and lay a solid foundation for sustainable social development. This article uses PVAR and threshold models to empirically study the rural revitalization effect under the interaction between tourism-driven poverty reduction and agro-tourism integration, as well as the dynamic correlation and threshold effect among the three. The following conclusions are drawn:

(1) Over the study period, rural revitalization and agro-tourism integration increased consistently, while tourism-driven poverty reduction exhibited a more complex trend. Spatially, rural revitalization had a central > eastern > northeastern > western pattern, tourism-driven poverty reduction efficiency followed a central > western > eastern > northeastern pattern, and agro-tourism integration evolved from a disorderly stage to a coordinated one.

(2) There were significant dynamic relationships among tourism-driven poverty reduction, agro-tourism integration, and rural revitalization. They exhibited bidirectional Granger causality, with rural revitalization influencing tourism-driven poverty reduction and vice versa. The explanatory power of both tourism-driven poverty reduction and agro-tourism integration in rural revitalization increased over time, with tourism-driven poverty reduction positively responding to agro-tourism integration. Improving tourism-driven poverty reduction efficiency promoted agro-tourism integration and, consequently, rural revitalization.

(3) The tourism-driven poverty reduction acted as an intermediate factor in the agro-tourism integration process for rural revitalization. When below a certain threshold, agro-tourism integration significantly promoted rural revitalization. After exceeding this threshold, agro-tourism integration's role in advancing rural revitalization intensified.

(4) The rural revitalization was influenced by various factors. Positive effects were observed from economic development levels, industrial structure, road density, financial support for agriculture, and human capital. Conversely, urbanization rates and the extent of regional openness to the outside world had negative impacts. The agricultural structure also played a role but was not statistically significant. Rural revitalization is a systematic project; therefore, efforts need to be made from various aspects.

### 6.3. Deficiencies and Prospects

This paper clarified the dynamic interaction between agro-tourism integration, tourism-driven poverty reduction, and rural revitalization, and based on the logical relationship that the improvement in tourism poverty reduction efficiency can promote the further development of agro-tourism integration, which in turn can promote rural revitalization, a threshold effect analysis was conducted to determine at what level of tourism-driven poverty reduction efficiency, agro-tourism integration can more effectively promote rural revitalization, but there still exist some shortcomings in the data collection and research content. In this paper, data from 31 provinces in mainland China were selected for analysis, but the development level of agro-tourism integration, tourism-driven poverty reduction, and rural revitalization varies greatly among provinces, so in the future, specific provinces

can be selected for in-depth analysis by combining case studies and empirical analysis. In addition, this paper is based on the provincial level and does not consider the spatial correlation of the development of rural revitalization in each province, and the spatial spillover effects among the three can be further explored by using spatial econometric models.

**Author Contributions:** Conceptualization, D.M.; methodology, D.M.; software, D.M.; validation, D.S.; formal analysis, D.M.; resources, Z.W. and D.S.; data curation, D.M.; writing—original draft preparation, M.D; writing—review and editing, Z.W. and D.S.; visualization, M.D; supervision, Z.W. and D.S.; funding acquisition, Z.W. and D.S. All authors have read and agreed to the published version of the manuscript.

**Funding:** This study was supported by the National Natural Science Foundation of China (41971162) and the Postgraduate Research & Practice Innovation Program of Jiangsu Province (KYCX23_1533).

**Institutional Review Board Statement:** Not applicable.

**Informed Consent Statement:** Not applicable.

**Data Availability Statement:** Data are contained within the article.

**Conflicts of Interest:** The authors declare no conflicts of interest.

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
