# Peer review of "Exploring the Rural Revitalization Effect under the Interaction of Agro-Tourism Integration and Tourism-Driven Poverty Reduction: Empirical Evidence for China"

_land, doi:10.3390/land13010060_

Round 1

Reviewer 1 Report

Comments and Suggestions for Authors

This paper constructs an analytical framework that includes tourism-driven poverty reduction, agro-tourism integration, and rural revitalization. Through empirical testing of the rural revitalization effect and their dynamic relationship under the interaction between a agro-tourism integration and tourism-driven poverty reduction, it is found that agro-tourism integration and tourism-driven poverty reduction can have a profound impact on China's economy, society, ecology, and other aspects from the perspective of tourism, enriching the connotation of rural economy, It helps to promote the coordinated development of urban and rural areas and the sustainable development of the tourism industry, and has certain theoretical and methodological significance. However, there are still the following issues that need further modification:

1. As the facade of a paper, the purpose of a paper abstract is to use concise text to describe the general content of the literature, so that readers can obtain necessary basic information without reading the entire text. Regardless of the structure of the abstract, it should try to include five elements: problem statement, research motivation, research methods, main results, and research conclusions. However, the text organization of the abstract is relatively simple and lacks necessary content.

2. In the introduction section, the author introduces the world and China development background of the research topic and the significance of the research topic, but lacks a description of the innovative points of the article. This article focuses on studying Chinese cases, hoping that in addition to case studies, the paper can also have a broader impact on tourism destinations.

3. Some maps in the paper are not standardized. Please refer to the requirements of the journal for revision to improve the accuracy and clarity of the images. Carefully check the format of the article and pay attention to the use of punctuation marks.

4. I suggest you check all of your tables' formats. They are a little bit different from the template.

5. In the discussion section, the author conducted in-depth discussions based on empirical results. The author pointed out that "How, rural revitalization is a comprehensive ending requirement collective effects from various sectors." However, there was no in-depth discussion on which aspects China should exert efforts to promote further development of rural revitalization. Please add relevant content.

6. It is recommended to add the Deficiencies and Prospects section. The article selects provinces in China as the research object. However, due to the vast territory of China, there are significant differences in the agro-tourism integration, tourism-driven poverty reduction, and rural revitalization development levels among different regions. Therefore, there may be limitations in data collection. Please provide necessary explanations and prospects around relevant topics to provide necessary ideas for future research.

Comments on the Quality of English Language

none

Author Response

Dear Editor,

Thank you and reviewers very much for your and reviewers’ critical comments on this manuscript entitled “Exploring the rural revitalization effect under the interaction of agro-tourism integration and tourism-driven poverty reduction : empirical evidence for China” (Manuscript ID: land-2755734).

We would like to thank the editor and all reviewers for their interest in our paper and for their valuable comments and suggestions to improve the quality of the manuscript. With that guidance, we have addressed the comments raised by the reviewers, and would like to re-submit it for your consideration. Our detailed responses to the reviewers’ comments and suggestions are listed as follows:

Thank you very much for your efforts to enhance the quality of our paper.

If there appears any question, please do not hesitate to contact me.

Yours sincerely,

Reviewer 2 Report

Comments and Suggestions for Authors

- In the Abstract, present more precisely what the goal of the research was.

- Also, in the Abstract, state the methods used in the research. List the methods immediately before the key findings.

- It is not clear why hypotheses are given in this way: 1a, 1b, etc. It is enough to state the main or null hypothesis and one or more sub-hypotheses or show one or more equal hypotheses.

- Is the interpolation method, which is used to a lesser extent in this research, reliable?

- Why are the data until 2020? Is there a possibility to use the data until 2022?

- Move the discussion (6.2) after the results or incorporate it together with the research results, which was partially done. No Discussion can come after the Conclusion. After the Conclusion, policy recommendations, study limitations, and future research directions can be discussed. The request is that the Discussion section (6.2: first paragraph) be broken up / moved before the Results Analysis section and slightly improved, where it is important to compare with the results of other authors. The remaining two paragraphs from section 6.2. modify and incorporate into the Conclusion.

- The first paragraph in the Conclusion is not needed, and it should be deleted. It has already been presented.

- It was a pleasure for me to read the manuscript and learn more about the topic.

- Good luck!

Comments on the Quality of English Language

/

Author Response

Response to the Editor and Reviewers’ Comments

To Reviewer #2

Q1: In the Abstract, present more precisely what the goal of the research was. Also, in the Abstract, state the methods used in the research. List the methods immediately before the key findings.

Response to Reviewer comment No.1: Thanks for your valuable comment. We have realized the importance of abstract writing. In the revised manuscript, we have reorganized the abstract according to the reviewer's requirements and highlighted the research objectives and methods of this study. The specific content is as follows: “Under the robust impetus of China's rural revitalization strategy, the agro-tourism integration and tourism-driven poverty reduction have profoundly impacted various aspects of China's economy, society, and ecology. This has propelled coordinated urban-rural development and the sustainable development of the tourism industry. This study introduces an analytical framework encompassing tourism-driven poverty reduction, the agro-tourism integration, and rural revitalization. Through PVAR and threshold models, it empirically examines the interactive effects, dynamic relationships, and threshold effects between the agro-tourism integration, tourism-driven poverty reduction, and rural revitalization. The conclusions are as follows: First, the indices of rural revitalization and the level of agro-tourism integration show an increasing trend across Chinese provinces, with varying trends in tourism-driven poverty reduction efficiency. Second, there is a significant dynamic rela-tionship among these factors, with the explanatory power of tourism-driven poverty reduction and agro-tourism integration gradually strengthening. The agro-tourism integration is identified as the most effective means of driving rural revitalization, while tourism-driven poverty reduction has a relatively weaker direct impact. Third, tourism-driven poverty reduction exhibits a positive impulse response to agro-tourism integration. The improvement in tourism-driven poverty reduction effi-ciency propels further development in agro-tourism integration, thereby fostering rural revitaliza-tion. The efficiency of tourism-driven poverty reduction presents a single threshold effect in the process of agro-tourism integration promoting rural revitalization. Fourth, the development of China's tourism industry has become an indispensable means of promoting rural revitalization and poverty reduction. However, rural revitalization is a comprehensive project influenced by various factors, requiring improvements and development across multiple aspects.” (lines 13-33)

Q2: It is not clear why hypotheses are given in this way: 1a, 1b, etc. It is enough to state the main or null hypothesis and one or more sub-hypotheses or show one or more equal hypotheses.

Response to Reviewer comment No.2: Thanks for your constructive comments. In the revised manuscript, we have reorganized the writing of the hypothesis section according to the reviewer's requirements, replacing the 1a and 1b forms with 1-6 forms, which further reflects the equally important parallelism of the hypotheses in this article. (lines 252-319)

Q3: Is the interpolation method, which is used to a lesser extent in this research, reliable? Why are the data until 2020? Is there a possibility to use the data until 2022?

Response to Reviewer comment No.3: Thanks for your valuable comment. Due to the missing data of individual years in the China Statistical Yearbook, we have supplemented some missing data with interpolation method to ensure the smooth progress of experimental models and methods. Interpolation method is a process or method of inferring new data points within a range from known and discrete data points, commonly used in the fields of mathematics and engineering. Due to the limited amount of missing data, using interpolation methods will not have a significant impact on the overall results. In addition, due to the fact that individual indicator data in the two major indicator systems of tourism poverty reduction and agricultural tourism integration have only been collected by countries up to 2020, in order to ensure the accuracy of the research and the authenticity and reliability of the data, we have set the research interval to 2020. After updating relevant indicators or conducting other related research, the data can still be expanded and used up to 2022.

Q4: Move the discussion (6.2) after the results or incorporate it together with the research results, which was partially done. No Discussion can come after the Conclusion. After the Conclusion, policy recommendations, study limitations, and future research directions can be discussed. The request is that the Discussion section (6.2: first paragraph) be broken up / moved before the Results Analysis section and slightly improved, where it is important to compare with the results of other authors. The remaining two paragraphs from section 6.2. modify and incorporate into the Conclusion..

Response to Reviewer comment No.4: Thanks for your constructive comments. In the revised draft, we have made modifications to the Conclusion and Discussion section as required. We have moved the Discussion section forward, reorganized the Conclusion, and added the Definitions and Prospects section. The specific content is as follows:

"6.1 Discussion

With the booming development of tourism in China, tourism has become an in-dispensable tool for promoting rural revitalization and poverty reduction. However, the direct contribution of tourism-driven poverty reduction to rural revitalization in China is still weak, probably due to the unbalanced development of China's regions, which makes tourism resources unevenly distributed, and the efficiency of tour-ism-driven poverty reduction varies widely. Although some regions are rich in tourism resources, the development of tourism is limited due to problems such as inconvenient transportation and poor infrastructure, making it difficult to fully play a role in pov-erty reduction. In addition, the traditional forms of tourism in most areas of China are still deeply rooted and lack innovation, transformation, and upgrading to meet the changing needs of the tourism market and the new consumer demands of tourists. Therefore, there is an urgent need to promote innovation, transformation, and up-grading of tourism in rural areas to improve the added value and competitiveness of tourism, and the integration of agriculture and tourism is the most effective way to do this. Through agro-tourism integration, we can effectively promote the transformation of rural economic structure, make the rural economy more diversified, promote the development of rural tourism and agriculture as well as the transformation of rural economic structure, increase the economic income of rural areas, improve the effi-ciency of tourism-driven poverty reduction, and in turn promote rural revitalization.

Furthermore, under the strong impetus of China's rural revitalization strategy, the integration of agriculture and tourism can provide a fresh perspective on the cultural value of agriculture. It highlights the contemporary significance and comprehensive benefits of agricultural culture, allowing it to blend with modern life from a tourist's viewpoint. Throughout the entire tourism process, agricultural culture is imprinted, making it more credible, tangible, and transmissible. The deep integration of agricul-ture and tourism in China allows advanced elements of the tourism industry to take root in various agricultural processes. Additionally, the tourism industry's reliance on ecological environments triggers a reassessment and amplification of the ecological value of agriculture. This contributes to enhancing agricultural ecological efficiency, opening new opportunities for China's agricultural modernization while promoting sustainable tourism development. However, rural revitalization is a systematic project that requires the joint efforts of all aspects of society, so China should also make efforts to promote rural revitalization from the following aspects: Improving rural roads, railroads and communications infrastructure, improving transportation, information and other infrastructure; Improving the rural financial system, expanding the cover-age of rural financial services, and increasing the convenience of financing for farmers; Integrating the coordinated development of urban and rural areas and promoting the optimal allocation of various resources; Promoting further opening of rural areas, strengthening scientific and technological innovation and technology introduction, encouraging and supporting rural enterprises to go to the international market, and improving the international competitiveness of related products; Strengthening hu-man resources training, improving farmers' skills and quality through various means, vigorously cultivating new agricultural subjects, optimizing subsidy policies, and at-tracting college students and other highly qualified personnel to come to work and start businesses.

6.2 Conclusion

As two important means to promote rural revitalization, tourism-driven poverty reduction and agro-tourism integration are in line with the requirements of the United Nations Sustainable Development Goals (SDGs). By promoting the development of tourism in rural areas, they provide local residents with ways to increase income and employment opportunities, effectively alleviate poverty, highlight the principle of so-cial responsibility, and lay a solid foundation for sustainable social development. This article uses PVAR and threshold models to empirically study the rural revitalization effect under the interaction between tourism-driven poverty reduction and agro-tourism integration, as well as the dynamic correlation and threshold effect among the three. The following conclusions are drawn:

(1) Over the study period, rural revitalization and agro-tourism integration in-creased consistently, while tourism-driven poverty reduction exhibited a more com-plex trend. Spatially, rural revitalization had a central > eastern > northeastern > western pattern, tourism-driven poverty reduction efficiency followed a central > western > eastern > northeastern pattern, and agro-tourism integration evolved from a disorderly stage to a coordinated one.

(2) There were significant dynamic relationships among tourism-driven poverty reduction, agro-tourism integration, and rural revitalization. They exhibited bidirec-tional Granger causality, with rural revitalization influencing tourism-driven poverty reduction and vice versa. The explanatory power of both tourism-driven poverty re-duction and agro-tourism integration in rural revitalization increased over time, with tourism-driven poverty reduction positively responding to agro-tourism integration. Improving tourism-driven poverty reduction efficiency promoted agro-tourism inte-gration and, consequently, rural revitalization.

(3) The tourism-driven poverty reduction acted as an intermediate factor in the agro-tourism integration process for rural revitalization. When below a certain threshold, agro-tourism integration significantly promoted rural revitalization. After exceeding this threshold, agro-tourism integration's role in advancing rural revitaliza-tion intensified.

(4) The rural revitalization was influenced by various factors. Positive effects were observed from economic development levels, industrial structure, road density, financial support for agriculture, and human capital. Conversely, urbanization rates and the extent of regional openness to the outside world had negative impacts. The ag-ricultural structure also played a role but was not statistically significant. Rural revi-talization is a systematic project, therefore efforts need to be made from various as-pects.". (lines 784-970)

Q5: The first paragraph in the Conclusion is not needed, and it should be deleted. It has already been presented.

Response to Reviewer comment No.5: Thanks for your valuable comment. In the revised draft, we have made modifications to the Conclusion section as required. We have deleted the first sentence and reorganized the conclusion. The specific content is as follows: " As two important means to promote rural revitalization, tourism-driven poverty reduction and agro-tourism integration are in line with the requirements of the United Nations Sustainable Development Goals (SDGs). By promoting the development of tourism in rural areas, they provide local residents with ways to increase income and employment opportunities, effectively alleviate poverty, highlight the principle of so-cial responsibility, and lay a solid foundation for sustainable social development. This article uses PVAR and threshold models to empirically study the rural revitalization effect under the interaction between tourism-driven poverty reduction and agro-tourism integration, as well as the dynamic correlation and threshold effect among the three. The following conclusions are drawn:

(1) Over the study period, rural revitalization and agro-tourism integration in-creased consistently, while tourism-driven poverty reduction exhibited a more com-plex trend. Spatially, rural revitalization had a central > eastern > northeastern > western pattern, tourism-driven poverty reduction efficiency followed a central > western > eastern > northeastern pattern, and agro-tourism integration evolved from a disorderly stage to a coordinated one.

(2) There were significant dynamic relationships among tourism-driven poverty reduction, agro-tourism integration, and rural revitalization. They exhibited bidirec-tional Granger causality, with rural revitalization influencing tourism-driven poverty reduction and vice versa. The explanatory power of both tourism-driven poverty re-duction and agro-tourism integration in rural revitalization increased over time, with tourism-driven poverty reduction positively responding to agro-tourism integration. Improving tourism-driven poverty reduction efficiency promoted agro-tourism inte-gration and, consequently, rural revitalization.

(3) The tourism-driven poverty reduction acted as an intermediate factor in the agro-tourism integration process for rural revitalization. When below a certain threshold, agro-tourism integration significantly promoted rural revitalization. After exceeding this threshold, agro-tourism integration's role in advancing rural revitaliza-tion intensified.

(4) The rural revitalization was influenced by various factors. Positive effects were observed from economic development levels, industrial structure, road density, financial support for agriculture, and human capital. Conversely, urbanization rates and the extent of regional openness to the outside world had negative impacts. The ag-ricultural structure also played a role but was not statistically significant. Rural revi-talization is a systematic project, therefore efforts need to be made from various as-pects." (lines 937-970)

Reviewer 3 Report

Comments and Suggestions for Authors

The article is clear and follows a line harmonized with its topic. My only concern would be that they delve into the bibliographic and theoretical review of the concept of pluriactive territories, which could even help to better understand the conclusions that are clear, but that leave a gap in the initial question about whether the articulation between tourism and agritourism is enough to alleviate rural poverty. The methodology used and the results are very clear and demonstrate correct use of the data obtained. Finally, just as additional information to the object of study, it would be important to answer the question, have state policies done enough to develop this tourism model?

Author Response

Response to the Editor and Reviewers’ Comments

To Reviewer #3

Q1: The article is clear and follows a line harmonized with its topic. My only concern would be that they delve into the bibliographic and theoretical review of the concept of pluriactive territories, which could even help to better understand the conclusions that are clear, but that leave a gap in the initial question about whether the articulation between tourism and agritourism is enough to alleviate rural poverty. The methodology used and the results are very clear and demonstrate correct use of the data obtained. Finally, just as additional information to the object of study, it would be important to answer the question, have state policies done enough to develop this tourism model?

Response to Reviewer comment No.1: Thanks for your valuable comment. In addition, we also express our gratitude to the reviewers for their affirmation of our article, but we still realize that we still have some shortcomings that need improvement. We have carefully read and revised the revision comments of other reviewers.

Next, we will answer your question, question one: "Why the article between tourism and aggression is enough to allocate rural power?"

According to existing research, the combination of agriculture and tourism can effectively promote the development of rural poverty reduction. The literature review section of this study also provides detailed explanations and summaries. In addition, the integration of agriculture and tourism refers to a model that combines agriculture and tourism to promote agricultural economic growth and increase farmers' income through the development of rural tourism. As a new mode of tourism development, agriculture tourism integration has received wide attention and discussion. For most rural areas, agriculture tourism integration is becoming an effective means to realize the transformation and upgrading of rural economy, promote the development of rural industries and solve the problems of agricultural development, providing a new way of development for the integration of three industries in rural areas. The integration of agriculture and tourism can attract tourists to rural areas for consumption and increase farmers' income by developing rural tourism attractions and rural leisure projects; Developing agricultural product processing industry, increasing added value, expanding market sales channels, and increasing farmers' income; Train farmers to engage in tourism service work, such as tour guides, homestay management, etc., provide employment opportunities and increase sources of income; Explore and protect cultural heritage and ecological resources in rural areas, create distinctive rural tourism products, attract tourists, and create economic benefits; Develop relevant policies, provide financial support and tax incentives, guide and promote the integrated development of agriculture and tourism, and promote poverty alleviation in rural areas. It should be noted that the success of agricultural tourism integration also needs to consider factors such as market demand, infrastructure construction, ecological environment protection, and farmer training.

question two: " have state policies done enough to develop this tourism model?"

The tourism industry, as an important pillar industry for rural revitalization, is not only an important component of national economic development, but also an important way to promote poverty alleviation. For example, fully utilizing tea resources and promoting tourism through tea in Neiyang Village, Fujian; Mount Taishan Village in Zhengzhou, which introduces cultural and tourism brand Qianjiaji and develops rural tourism with characteristics; Shandong Zhuquan Village, which utilizes the idyllic beauty of the village and the unique Yimeng rural culture to develop a rural leisure tourism industry with Yimeng characteristics, and so on, all of these villages use tourism as a means of poverty alleviation and development. For example, in Zhonghaoyu Village, Zibo City, Shandong Province, the "Zhonghaoyu Village Tourism Company" was established, implementing a collective operation model for rural tourism, continuously developing and improving the tourism management model, and steadily advancing the growth rate of tourist numbers. From a poverty-stricken village with an average annual income of less than 2000 yuan 20 years ago, it has entered the ranks of affluent villages; Following the development concept of "interaction between the three industries and integration of agriculture and tourism", highlighting the theme of red culture, and relying on modern agriculture to develop rural tourism in the new rural areas of Baiping Feilong in Sichuan Province; Hongsha Village in Sansheng Township, Chengdu, Sichuan Province, utilizes characteristic flower planting to renovate and renovate the rural residential areas, attracting a large number of tourists to visit. It has become the preferred destination for sightseeing and tourism in the suburbs of Chengdu, greatly promoting the coordinated development of urban and rural areas in Chengdu; By showcasing Miao culture and relying on rich ethnic cultural resources, we have won honors such as the National Agricultural Tourism Demonstration Site and the Best Folk Culture Award of China Rural Tourism 'Flying Swallow Award', which has increased the income of local farmers and improved their living standards. In the past 30 years, tourism poverty reduction has risen from local exploration to a national strategy, and has developed from specific practices in individual regions to nationwide universal demonstrations, making tremendous contributions to the reduction of poverty in China and even the global anti poverty cause.

Reviewer 4 Report

Comments and Suggestions for Authors

Study  examines the period from 2006 to 2020 across China's 31 provinces. It assesses the levels of agro-tourism integration, tourism-driven poverty reduction, and rural  revitalization using the PVAR and threshold models. The research explores how agro- tourism integration and tourism-driven poverty reduction affect rural revitalization and their dynamic interactions.

Author Response

Response to the Editor and Reviewers’ Comments

To Reviewer #4

Q1: Study examines the period from 2006 to 2020 across China's 31 provinces. It assesses the levels of agro-tourism integration, tourism-driven poverty reduction, and rural revitalization using the PVAR and threshold models. The research explores how agro- tourism integration and tourism-driven poverty reduction affect rural revitalization and their dynamic interactions.

Response to Reviewer comment No.1: Thanks for your valuable comment. In addition, we also express our gratitude to the reviewers for their affirmation of our article, but we still realize that we still have some shortcomings that need improvement. We have carefully read and revised the revision comments of other reviewers. Under the robust impetus of China's rural revitalization strategy, the agro-tourism integration and tourism-driven poverty reduction have profoundly impacted various aspects of China's economy, society, and ecology. This has propelled coordinated urban-rural development and the sustainable development of the tourism industry. This study introduces an analytical framework encompassing tourism-driven poverty reduction, the agro-tourism integration, and rural revitalization. Through PVAR and threshold models, it empirically examines the interactive effects, dynamic relationships, and threshold effects between the agro-tourism integration, tourism-driven poverty reduction, and rural revitalization. The conclusions are as follows: First, the indices of rural revitalization and the level of agro-tourism integration show an increasing trend across Chinese provinces, with varying trends in tourism-driven poverty reduction efficiency. Second, there is a significant dynamic rela-tionship among these factors, with the explanatory power of tourism-driven poverty reduction and agro-tourism integration gradually strengthening. The agro-tourism integration is identified as the most effective means of driving rural revitalization, while tourism-driven poverty reduction has a relatively weaker direct impact. Third, tourism-driven poverty reduction exhibits a positive impulse response to agro-tourism integration. The improvement in tourism-driven poverty reduction effi-ciency propels further development in agro-tourism integration, thereby fostering rural revitaliza-tion. The efficiency of tourism-driven poverty reduction presents a single threshold effect in the process of agro-tourism integration promoting rural revitalization. Fourth, the development of China's tourism industry has become an indispensable means of promoting rural revitalization and poverty reduction. However, rural revitalization is a comprehensive project influenced by various factors, requiring improvements and development across multiple aspects.

Reviewer 5 Report

Comments and Suggestions for Authors

Please, see the attached document.

Author Response

Response to the Editor and Reviewers’ Comments

To Reviewer #5

Q1: Title:The title still does not reflect that the research presents a case study, which may lead to misunderstandings.

Response to Reviewer comment No.1: Thanks for your valuable comment. Our study is based on the rapid development of China's rural tourism industry and the promotion of rural revitalization development strategy. From the perspective of industry integration and tourism poverty reduction interaction, we take 31 provinces (cities, districts) in China from 2006 to 2020 as the research object to measure their level of agricultural tourism integration, tourism poverty reduction efficiency, and rural revitalization development index, And use PVAR and threshold models to empirically study the rural revitalization effect under the interaction between agricultural tourism integration and tourism poverty reduction, as well as the dynamic correlation and threshold effect between the three. The title of the paper is “Exploring the rural revitalization effect under the interaction of agro-tourism integration and tourism-driven poverty reduction : empirical evidence for China”. It is a case study of 31 provinces in China, which is clearly reflected in the title. (lines 2-4)

Q2: Abstract:The maximum number of words recommended by the magazine continues to be exceeded.

Response to Reviewer comment No.2: Thanks for your constructive comments. After referring to the published articles and requirements of this journal, and taking into account the opinions of other reviewers, we have reorganized the abstract section. The specific content is " Under the robust impetus of China's rural revitalization strategy, the agro-tourism integration and tourism-driven poverty reduction have profoundly impacted various aspects of China's economy, society, and ecology. This has propelled coordinated urban-rural development and the sustainable development of the tourism industry. This study introduces an analytical framework encompassing tourism-driven poverty reduction, the agro-tourism integration, and rural revitalization. Through PVAR and threshold models, it empirically examines the interactive effects, dynamic relationships, and threshold effects between the agro-tourism integration, tourism-driven poverty reduction, and rural revitalization. The conclusions are as follows: First, the indices of rural revitalization and the level of agro-tourism integration show an increasing trend across Chinese provinces, with varying trends in tourism-driven poverty reduction efficiency. Second, there is a significant dynamic rela-tionship among these factors, with the explanatory power of tourism-driven poverty reduction and agro-tourism integration gradually strengthening. The agro-tourism integration is identified as the most effective means of driving rural revitalization, while tourism-driven poverty reduction has a relatively weaker direct impact. Third, tourism-driven poverty reduction exhibits a positive impulse response to agro-tourism integration. The improvement in tourism-driven poverty reduction effi-ciency propels further development in agro-tourism integration, thereby fostering rural revitaliza-tion. The efficiency of tourism-driven poverty reduction presents a single threshold effect in the process of agro-tourism integration promoting rural revitalization. Fourth, the development of China's tourism industry has become an indispensable means of promoting rural revitalization and poverty reduction. However, rural revitalization is a comprehensive project influenced by various factors, requiring improvements and development across multiple aspects." (lines 13-33)

Q3: Keywords:There are still matches between the keywords and those in the article title.

Response to Reviewer comment No.3: Thanks for your valuable comment. Keywords are the literature identifiers of scientific papers, which are natural language vocabulary used to express the thematic concepts of the literature. Keywords are words or phrases selected from the title, abstract, hierarchical title, or main text of a paper that have key physical meanings and can reflect the main concepts of the paper. After careful consideration, this article has finally identified the following 5 keywords " agro-tourism integration; tourism-driven poverty reduction; rural revitalization; PVAR model; threshold model ". (lines 34-35)

Q4: Theoretical framework - references:

The theoretical framework has not yet been enriched with the contributions of authors

beyond Chinese literature. Land is a prestigious international magazine; the article addresses a subject widely studied, from various perspectives, by authors of many other nationalities. This remains a shortcoming of the article..

Response to Reviewer comment No.4: Thanks for your constructive comments. The development of China's rural tourism industry plays an important role in promoting the rural revitalization strategy. At the same time, with the strong promotion of China's rural revitalization strategy, the integration of agriculture and tourism and poverty reduction through tourism have also had a profound impact on many aspects of China's economy, society, ecology, etc. from the perspective of tourism, enriching the connotation of rural economy, promoting coordinated urban-rural development, and sustainable development of the tourism industry. This study constructs for the first time an analytical framework that includes tourism poverty reduction, agricultural tourism integration, and rural revitalization. Through PVAR and threshold models, it empirically tests the rural revitalization effect under the interaction between agricultural tourism integration and tourism poverty reduction, as well as their dynamic relationship and threshold effect.

Q5: Results of the investigation:

  • A better definition of the research objectives is still necessary to identify them with the content of these subsections.
  • A careful re-reading of them with the aim of making an important synthesis is still not carried out.

Response to Reviewer comment No.5: Thanks for your valuable comment. In the revised manuscript, we have made it clearer that the research objective of this article is to use the entropy method, PVAR model, and threshold model to analyze whether tourism poverty reduction and agricultural tourism integration can promote rural revitalization, as well as the interrelationships among the three, with the aim of enriching the research scope and theoretical system of relevant aspects and deepening our understanding of the relationship between the three, Provide Chinese experience for the development of rural tourism and rural revitalization construction in other countries. And the relevant research topics have been defined in the form of literature review and introduction.

Q6: Discussion:

The document still does not have a discussion section; The incorporation of the term

discussion in the name of the fourth section does not remedy this lack.

Conclusions:

  • It is still necessary to relate the conclusions to the research objectives and

contextualize them with the different counties that are part of the analyzed region. In

their current version, these conclusions continue to detract from the statistical effort

made.

  • The rest of the paragraphs lack scientific support and seem like recommendations

from the authors beyond the content of the research. I would seriously consider

incorporating it into the document; at least, in the conclusions section: worded

differently, perhaps, in the discussion section, they could be presented with due

reference to other authors who defend those same theses.

Response to Reviewer comment No.6: Thanks for your constructive comments. In the revised draft, we have made modifications to the Conclusion and Discussion section as required. We have moved the Discussion section forward, reorganized the Conclusion, and added the Definitions and Prospects section. The specific content is as follows:

"6.1 Discussion

With the booming development of tourism in China, tourism has become an in-dispensable tool for promoting rural revitalization and poverty reduction. However, the direct contribution of tourism-driven poverty reduction to rural revitalization in China is still weak, probably due to the unbalanced development of China's regions, which makes tourism resources unevenly distributed, and the efficiency of tour-ism-driven poverty reduction varies widely. Although some regions are rich in tourism resources, the development of tourism is limited due to problems such as inconvenient transportation and poor infrastructure, making it difficult to fully play a role in pov-erty reduction. In addition, the traditional forms of tourism in most areas of China are still deeply rooted and lack innovation, transformation, and upgrading to meet the changing needs of the tourism market and the new consumer demands of tourists. Therefore, there is an urgent need to promote innovation, transformation, and up-grading of tourism in rural areas to improve the added value and competitiveness of tourism, and the integration of agriculture and tourism is the most effective way to do this. Through agro-tourism integration, we can effectively promote the transformation of rural economic structure, make the rural economy more diversified, promote the development of rural tourism and agriculture as well as the transformation of rural economic structure, increase the economic income of rural areas, improve the effi-ciency of tourism-driven poverty reduction, and in turn promote rural revitalization.

Furthermore, under the strong impetus of China's rural revitalization strategy, the integration of agriculture and tourism can provide a fresh perspective on the cultural value of agriculture. It highlights the contemporary significance and comprehensive benefits of agricultural culture, allowing it to blend with modern life from a tourist's viewpoint. Throughout the entire tourism process, agricultural culture is imprinted, making it more credible, tangible, and transmissible. The deep integration of agricul-ture and tourism in China allows advanced elements of the tourism industry to take root in various agricultural processes. Additionally, the tourism industry's reliance on ecological environments triggers a reassessment and amplification of the ecological value of agriculture. This contributes to enhancing agricultural ecological efficiency, opening new opportunities for China's agricultural modernization while promoting sustainable tourism development. However, rural revitalization is a systematic project that requires the joint efforts of all aspects of society, so China should also make efforts to promote rural revitalization from the following aspects: Improving rural roads, railroads and communications infrastructure, improving transportation, information and other infrastructure; Improving the rural financial system, expanding the cover-age of rural financial services, and increasing the convenience of financing for farmers; Integrating the coordinated development of urban and rural areas and promoting the optimal allocation of various resources; Promoting further opening of rural areas, strengthening scientific and technological innovation and technology introduction, encouraging and supporting rural enterprises to go to the international market, and improving the international competitiveness of related products; Strengthening hu-man resources training, improving farmers' skills and quality through various means, vigorously cultivating new agricultural subjects, optimizing subsidy policies, and at-tracting college students and other highly qualified personnel to come to work and start businesses.

6.2 Conclusion

As two important means to promote rural revitalization, tourism-driven poverty reduction and agro-tourism integration are in line with the requirements of the United Nations Sustainable Development Goals (SDGs). By promoting the development of tourism in rural areas, they provide local residents with ways to increase income and employment opportunities, effectively alleviate poverty, highlight the principle of so-cial responsibility, and lay a solid foundation for sustainable social development. This article uses PVAR and threshold models to empirically study the rural revitalization effect under the interaction between tourism-driven poverty reduction and agro-tourism integration, as well as the dynamic correlation and threshold effect among the three. The following conclusions are drawn:

(1) Over the study period, rural revitalization and agro-tourism integration in-creased consistently, while tourism-driven poverty reduction exhibited a more com-plex trend. Spatially, rural revitalization had a central > eastern > northeastern > western pattern, tourism-driven poverty reduction efficiency followed a central > western > eastern > northeastern pattern, and agro-tourism integration evolved from a disorderly stage to a coordinated one.

(2) There were significant dynamic relationships among tourism-driven poverty reduction, agro-tourism integration, and rural revitalization. They exhibited bidirec-tional Granger causality, with rural revitalization influencing tourism-driven poverty reduction and vice versa. The explanatory power of both tourism-driven poverty re-duction and agro-tourism integration in rural revitalization increased over time, with tourism-driven poverty reduction positively responding to agro-tourism integration. Improving tourism-driven poverty reduction efficiency promoted agro-tourism inte-gration and, consequently, rural revitalization.

(3) The tourism-driven poverty reduction acted as an intermediate factor in the agro-tourism integration process for rural revitalization. When below a certain threshold, agro-tourism integration significantly promoted rural revitalization. After exceeding this threshold, agro-tourism integration's role in advancing rural revitaliza-tion intensified.

(4) The rural revitalization was influenced by various factors. Positive effects were observed from economic development levels, industrial structure, road density, financial support for agriculture, and human capital. Conversely, urbanization rates and the extent of regional openness to the outside world had negative impacts. The ag-ricultural structure also played a role but was not statistically significant. Rural revi-talization is a systematic project, therefore efforts need to be made from various as-pects.". (lines 784-970)

Q7: Additional comment:

Beyond the introduction, the suggestions made in the first review report have not been substantially taken into account. On the contrary, the article now presents formal errors and oversights, which were not included in the first version, and which should not be part of a document at this stage of preparation. Some examples: lines 102 (repetition of the word limitations) or 112 (incorporation of Lei in red); lines 794-797 (blank); conclusions; bibliography (with the format changed).

Response to Reviewer comment No.7: Thanks for your valuable comment. In the revised manuscript, we have made the revisions as requested by the reviewer.

Round 2

Reviewer 5 Report

Comments and Suggestions for Authors

The observations made in the previous report have not been substantially taken into account by the authors. The response report omits some of them and, in some cases, the answers provided evade the issues raised.